# An Efficient Parallel Implementation of the Runge–Kutta Discontinuous Galerkin Method with Weighted Essentially Non-Oscillatory Limiters on Three-Dimensional Unstructured Meshes

**Weicheng Pei, Yuyan Jiang and Shu Li \***

School of Aeronautics Science and Engineering, Beihang University, 37 Xueyuan Road, Haidian District, Beijing 100191, China; weicheng.pei@icloud.com (W.P.); jiang.yu.yan@icloud.com (Y.J.)
\* Correspondence: lishu@buaa.edu.cn

**Abstract:** In computational fluid dynamics, high-order solvers suitable for three-dimensional unstructured meshes are attractive but are less developed than other methods. In this article, we provide the formulation and a parallel implementation of the Runge–Kutta discontinuous Galerkin finite element method with weighted essentially non-oscillatory limiters, which are compact and effective for suppressing numerical oscillations near discontinuities. In our experiments, high-order solvers do outperform their low-order counterparts in accuracy and the efficient parallel implementation makes the time cost affordable for large problems. Such high-order parallel solvers are efficient tools for solving conservative laws including the Euler system that models inviscid compressible flows.

**Keywords:** high-order CFD solvers; discontinuous Galerkin methods; WENO limiters; three-dimensional unstructured meshes; distributed memory parallelization

## 1. Introduction

Most of the computational fluid dynamics (CFD) solvers currently used in aerospace engineering are based on schemes using finite volume (FV) methods, which are more suitable than schemes using finite difference (FD) methods for unstructured meshes. However, FV schemes usually have only second-order spatial accuracy, due to the difficulty of handling irregular local stencils, whose sizes grows rapidly as the order of accuracy increases. In high-order FV schemes, the local stencil of a cell is made up of the cell and the cell's neighbors, and the neighbors' neighbors, and so on. To ease the development of high-order schemes for unstructured meshes, it is better to use finite element (FE) methods whose local stencils are much more compact than those of FV schemes. One family of such compact high-order schemes suitable for unstructured meshes are called the discontinuous Galerkin (DG) methods, which assume continuous approximation in each cell (like classic FE methods), but allow discontinuities to exist on cell boundaries (like classic FV methods). Such an assumption gives scheme designers more freedom on choosing the basis for each cell, such as orthogonal functions which lead to modal DG methods, and Lagrange polynomials which lead to nodal DG methods [1]. The original DG method was introduced by Reed and Hill [2] for solving the neutron transport equation, which is a linear problem contains only first-order spatial derivatives. For nonlinear conservation laws, Chavent and Salzano [3] made the first attempt of using a DG method to discretize in space and using the explicit Euler method for time discretization. This scheme is second-order accurate in space but only first-order accurate in time. To improve the accuracy of time discretization, Cockburn and Shu [4] replace the first-order Euler method with a special second-order Runge–Kutta (RK) method [5,6], which is explicit and total variation diminishing (TVD). This is the first successful Runge–Kutta discontinuous Galerkin (RKDG) method, which is high-order accurate both in space and time, for scalar conservation laws. This RKDG

method was soon extended to one-dimensional system case [7], multi-dimensional scalar case [8] and multi-dimensional system case [9], which includes the Euler system that models inviscid compressible flows. Nearly the same time (late 1990s), the DG methods were also extended to the Navier–Stokes system that governs viscous flows. Bassi and Rebay [10] made the first attempt to apply the DG idea to both the unknowns and their gradients for solving the Navier–Stokes equations. This method was then generalized by Cockburn and Shu [11] to the family of local discontinuous Galerkin (LDG) methods, which extends the RKDG methods from conversation laws to convection-dominated problems. One may refer to [12], which is a comprehensive review article on this topic. In the rest of this article, we will only study the RKDG methods for conservation laws that are purely hyperbolic. In Sections 2.1 and 2.3, we will give a matrix formed formulation of the RKDG method for solving three-dimensional conservation law systems.

When solving hyperbolic problems, limiting procedures, or limiters for short, are necessary for suppressing numerical oscillations that might occur near discontinuities (known as the Gibbs phenomenon [13]). For FD and FV schemes, there is the monotonic upstream-centered scheme for conservation laws (MUSCL) [14–17], which could achieve third-order accuracy when used with caution. However, it is not directly applicable to FE schemes. Besides the RKDG formulations, there is another important contribution made by [8,9], which is proposing the generalized slope limiters for multi-dimensional problems. These minmod type limiters, together with TVD RK methods, makes the high-order solutions free from non-physical oscillation, but tend to reduce the order of accuracy in smooth regions. To overcome this drawback, the essentially non-oscillatory (ENO) [18] and weighted ENO (WENO) [19] limiting procedures were introduced. Both of them can maintain high-order accuracy in smooth regions but essentially suppress spurious oscillations near discontinuities. The first attempt of making such limiters suitable for unstructured meshes was to incorporate a WENO reconstruction procedure into high-order FV schemes [20]. Similar ideas were later adopted for DG methods [21,22] at the cost of sacrificing the compactness of DG methods. Compact versions of WENO limiters suitable for DG methods only occurred in the last ten years [23–25] and most of them are only formulated for two-dimensional meshes. In Section 2.2, we will give a unified formulation of some compact WENO limiters for both two- and three-dimensional RKDG methods on unstructured meshes.

The two-dimensional version of the algorithms described in Sections 2.1–2.3 has been proposed for nearly ten years, but few three-dimensional or real engineering applications have been reported so far. One reason is that the amount of computational resources cost by each cell grows rapidly as the order of accuracy or the dimension of space increases. Fortunately, both RKDG methods and WENO limiters use highly compact stencils and there is no global algebraic equation to be solved (as in FD and FV schemes). Based on these facts, parallel computing using domain decomposition can be applied for accelerating computation in a very natural way. Currently, we have not seen any parallel implementation of RKDG methods with WENO limiters for three-dimensional unstructured meshes. However, there are some frequently referenced works existing in the literature, which measured the parallel efficiencies of their DG methods. Biswas, Devine and Flaherty [26] performed some tests of their one- and two-dimensional DG solvers for linear scalar problems on uniform structured meshes. They achieved excellent parallel efficiencies, which were over 99% for pure solution time (without I/O) and at least 89% for total running time. Bey, Patra and Oden [27] tested their DG solvers for linear conservation laws on both structured (uniform) and unstructured ($hp$-adaptive) meshes. They obtained nearly optimal speedups when the number of interior elements is sufficiently larger than that of subdomain boundary elements. Recently, Chalmers et al. [28] implemented their DG solver for two-dimensional Navier–Stokes equations using MPI/OpenMP hybrid parallelism and achieved good scalability on a uniform mesh with only quadrilateral elements. None of them show the parallel performance of DG solvers on three-dimensional unstructured meshes. In this article, we incorporate unstructured mesh partitioning and message passing into the algorithms and

implement them on top of publicly available libraries to support parallel execution. To partition a three-dimensional unstructured mesh, we use the application programming interface (API) provided by the METIS library [29]. To send and receive messages, we use the message passing interface (MPI) [30], which is the de facto industry standard of distributed memory parallelization. We will give the details of these parallel programming techniques in Section 2.4.

## 2. Methods

### 2.1. Spatial Discretization

The differential form of a conservation law system in a three-dimensional space could be written as

$$\partial_t \underline{U} + \partial_x \underline{F}^x + \partial_y \underline{F}^y + \partial_z \underline{F}^z = \underline{O} \tag{1}$$

in which

- $\partial_t, \partial_x, \partial_y, \partial_z$ represent $\frac{\partial}{\partial t}, \frac{\partial}{\partial x}, \frac{\partial}{\partial y}, \frac{\partial}{\partial z}$, respectively;
- $\underline{U}$ is a $K \times 1$ matrix of unknowns, each row of which is a scalar function depending on position $\vec{x}$ and time $t$;
- $\underline{F}^\mu$ (where $\mu = x, y, z$) is also a $K \times 1$ matrix, which is the dot-product of the flux $\underline{\vec{F}}$ (whose value depends on $\underline{U}$) and $\vec{e}_\mu$ (which is an unit vector along the positive direction of the $\mu$-axis);
- $\underline{O}$ is a $K \times 1$ matrix of 0's.

This differential equation could be turned into an integral equation, by multiplying both sides with an arbitrary function $V(\vec{x})$ and integrating the product on an arbitrary control volume $\Omega$:

$$\int_\Omega \left( \partial_t \underline{U} + \nabla \cdot \underline{\vec{F}} \right) V = \underline{O}.$$

To weaken the smoothness requirements on $\underline{\vec{F}}$, we apply integral by parts and Gauss's divergence theorem to it, which will lead to the weak form of Equation (1):

$$\int_\Omega \left( \partial_t \underline{U} - \underline{\vec{F}} \cdot \nabla V \right) + \int_{\partial\Omega} \left( \vec{v} \cdot \underline{\vec{F}} \right) V = \underline{O}, \tag{2}$$

where $\vec{v}$ is the outer normal unit vector of the control surface $\partial\Omega$ (which is the boundary of $\Omega$).

To introduce spatial discretization for Equation (2), we choose the linear space spanned by polynomials up to the $p$-th degree over $\Omega$, denoted as $\mathcal{V}^p(\Omega)$, as the approximation space. Let $\underline{\phi}(\vec{x}) = \left[ \phi_1(\vec{x}) \cdots \phi_L(\vec{x}) \right]$ be a basis of $\mathcal{V}^p(\Omega)$, in this article, we choose

$$\phi_1(\vec{x}) = 1, \qquad \begin{bmatrix} \phi_2(\vec{x}) \\ \phi_3(\vec{x}) \\ \phi_4(\vec{x}) \end{bmatrix} = \begin{bmatrix} x - x_0 \\ y - y_0 \\ z - z_0 \end{bmatrix}, \qquad \begin{bmatrix} \phi_5(\vec{x}) \\ \phi_6(\vec{x}) \\ \phi_7(\vec{x}) \\ \phi_8(\vec{x}) \\ \phi_9(\vec{x}) \\ \phi_{10}(\vec{x}) \end{bmatrix} = \begin{bmatrix} (x - x_0)(x - x_0) \\ (x - x_0)(y - y_0) \\ (x - x_0)(z - z_0) \\ (y - y_0)(y - y_0) \\ (y - y_0)(z - z_0) \\ (z - z_0)(z - z_0) \end{bmatrix}, \qquad \cdots,$$

in which $(x_0, y_0, z_0)$ is the geometric center of $\Omega$. Then $\underline{U}$ and $V$ could be approximated as

$$\underline{U}(\vec{x}, t) \approx \underline{U}^h(\vec{x}, t) = \sum_{l=1}^{L} \underline{\hat{U}}_l(t) \, \phi_l(\vec{x}), \qquad V(\vec{x}) \approx V^h(\vec{x}) = \sum_{l=1}^{L} \hat{V}_l \, \phi_l(\vec{x}),$$

where each $\underline{\hat{U}}_l(t)$ is a $K \times 1$ matrix of temporal functions, and each $\hat{V}_l$ is a constant number (which is arbitrary since $V(\vec{x})$ is arbitrary). Substitute them into the weak form (Equation (1)), we obtain

$$\sum_l \hat{V}_l \left[ \sum_k \left( \int_\Omega \phi_l \phi_k \right) \frac{d\hat{\underline{U}}_k}{dt} + \int_\Omega (\nabla \phi_l) \cdot \vec{\underline{F}} \left( \underline{U}^h \right) + \oint_{\partial\Omega} \phi_l \, \underline{F}^\nu \left( \underline{U}_I^h, \underline{U}_O^h \right) \right] = \underline{O} \qquad (3)$$

where $\vec{v} \cdot \vec{\underline{F}} =: \underline{F}^\nu$ is the normal flux on $\partial\Omega$, whose value could be solved from $\underline{U}_I^h$ (the approximated inner-side state) and $\underline{U}_O^h$ (the approximated outer-side state). We implement this procedure as an independent module called the Riemann solver of the conservation law (Equation (1)), see [31] for details.

Recall the arbitrariness of $\left\{ \hat{V}_l \right\}_{l=1}^L$ and adopt the inner-product notation

$$\langle f | g \rangle := \int_\Omega f(\vec{x}) \, g(\vec{x}),$$

we could turn Equation (3) into a system of ordinary differential equations:

$$\frac{d\hat{\underline{U}}_{K \times L}}{dt} = \underline{B}_{K \times L} \, \underline{A}_{L \times L}^{-1} =: \underline{R}_{K \times L} \qquad (4)$$

in which

$$\hat{\underline{U}}(t) = \begin{bmatrix} \langle U_1 | \phi_1 \rangle & \cdots & \langle U_1 | \phi_L \rangle \\ \vdots & \ddots & \vdots \\ \langle U_K | \phi_1 \rangle & \cdots & \langle U_K | \phi_L \rangle \end{bmatrix}_{K \times L}$$

is the matrix of temporal functions (which will be solved in Section 2.3), and

$$\underline{A} = \begin{bmatrix} \langle \phi_1 | \phi_1 \rangle & \cdots & \langle \phi_1 | \phi_L \rangle \\ \vdots & \ddots & \vdots \\ \langle \phi_L | \phi_1 \rangle & \cdots & \langle \phi_L | \phi_L \rangle \end{bmatrix}_{L \times L}$$

is a constant matrix for a given $\Omega$, and

$$\underline{B} = \int_\Omega \begin{bmatrix} \underline{F}^x & \underline{F}^y & \underline{F}^z \end{bmatrix}_{K \times 3} \begin{bmatrix} \partial_x \underline{\phi} \\ \partial_y \underline{\phi} \\ \partial_z \underline{\phi} \end{bmatrix}_{3 \times L} - \oint_{\partial\Omega} \underline{F}^\nu_{K \times 1} \, \underline{\phi}_{1 \times L} \qquad (5)$$

is a variable matrix depending on $\hat{\underline{U}}$, so is the residual matrix $\underline{R} = \underline{B} \, \underline{A}^{-1}$. By applying the Gram–Schmidt orthonormalization to $\phi$, the constant matrix $\underline{A}$ could be an identity matrix, which would lead to $\underline{R} = \underline{B}$. The integrals in $\underline{B}$ would be evaluated by Gaussian quadrature rules. For triangular and tetrahedral cells, we used the quadrature rules given in [32].

### 2.2. Limiting Procedures

The limiters we used in this article was originally designed by [23,24]. Zhong [23] gives the formulation for two-dimensional structured meshes, and Zhu [24] extends it for two-dimensional unstructured meshes. Here we present a unified formulation for both two- and three-dimensional unstructured meshes. To simplify subscripts, we denote $\psi|_{E_i}$ (the restriction of function $\psi$ on element $E_i$) as $\psi_i$, and $\langle \psi \rangle_{E_i}$ (the average value of $\psi$ on $E_i$) as $\langle \psi \rangle_i$. Let $K_i$ be the index set of $E_i$'s neighbors (those elements adjacent to $E_i$), and $K_i^+ := K_i \cup \{i\}$.

#### 2.2.1. The `ScalarWeno` Limiter

To reconstruct a scalar-valued function $\psi$ on $E_i$, we first borrow the expression of $\psi$ from $E_k$ to $E_i$

$$\psi_{k \to i}(\vec{x}) := \psi_k(\vec{x}) - \langle \psi_k \rangle_i + \langle \psi_i \rangle_i \qquad (6)$$

for each $k \in K_i^+$. The key idea of WENO limiters is to build a convex combination of these borrowed functions:

$$\psi_i^{\text{new}}(\vec{x}) := \sum_{k \in K_i^+} w_{k \to i}\, \psi_{k \to i}(\vec{x}). \tag{7}$$

The non-negative weight $w_{k \to i}$ should be determined from the smoothness of $\psi_{k \to i}$, so we then calculate the smoothness of $\psi_{k \to i}$ for each $k \in K_i$:

$$\beta_{k \to i} = \sum_{|\alpha|=1}^{p} \frac{l_i^{2|\alpha|}}{|E_i|} \int_{E_i} \left( \frac{\partial^{|\alpha|} \psi_{k \to i}}{\partial x_1^{\alpha_1} \cdots \partial x_d^{\alpha_d}} \right)^2, \qquad |\alpha| := \alpha_1 + \cdots + \alpha_d \tag{8}$$

in which $|E_i| := \int_{E_i} 1$ is the measure (i.e., "area" for $d = 2$, "volume" for $d = 3$) of $E_i$, and $l_i := \sqrt[d]{|E_i|}$ is the approximated length of $E_i$'s edges. Once we have these $\beta$'s, the weight for each $\psi_{k \to i}$ could be constructed as

$$w_{k \to i} = \frac{w_{k \to i}^{\beta}}{\sum_{k \in K_i^+} w_{k \to i}^{\beta}}, \qquad w_{k \to i}^{\beta} := \frac{w_{k \to i}^{\star}}{(\varepsilon_0 + \beta_{k \to i})^2}, \tag{9}$$

in which

$$w_{k \to i}^{\star} = \begin{cases} \varepsilon_1 & k \neq i \\ 1 - \sum_{j \in K_i} \varepsilon_1 & k = i \end{cases} \tag{10}$$

are called the ideal weights. The $\varepsilon$'s in Equations (9) and (10) are artificial parameters and we use $\varepsilon_0 = 10^{-6}$ and $\varepsilon_1 = 10^{-3}$ as suggested by [23,24]. This limiting procedure for scalar-valued functions is independent from the conservation laws to be solved, so it can be programmed as an independent module, which we would like to name as the `ScalarWeno` limiter.

### 2.2.2. The `EigenWeno` Limiter

For a conservation law system (Equation (1)), the value of the conservative variable $\underline{U}$ is a column matrix, for which the following limiter is recommended. The first step is to obtain the $\nu$-split form of Equation (1) on the interface shared by $E_i$ and its neighbor $E_k$ (for each $k \in K_i$):

$$\partial_t \underline{U} + \partial_\nu \underline{F}^\nu = \underline{O}, \tag{11}$$

where $\partial_\nu := \vec{v} \cdot \nabla$ is the directional derivative operator and $\underline{F}^\nu := \vec{v} \cdot \vec{\underline{F}}$ is the normal flux (as in Equation (3)). It can be treated as a one-dimensional conservation law system whose flux Jacobian can be approximated by the average value of $\underline{U}$:

$$\underline{A}^\nu = \left. \frac{\partial \underline{F}^\nu}{\partial \underline{U}} \right|_{\langle \underline{U} \rangle_i}. \tag{12}$$

For a hyperbolic system, it is guaranteed that the $K \times K$ matrix $\underline{A}^\nu$ has $K$ real eigenvalues and has the eigenvalue decomposition

$$\underline{A}^\nu = \underline{R} \begin{bmatrix} \lambda_1 & & \\ & \ddots & \\ & & \lambda_K \end{bmatrix} \underline{R}^{-1}, \qquad \underline{R} := \begin{bmatrix} \underline{r_1} & \cdots & \underline{r_K} \end{bmatrix}, \tag{13}$$

where $\underline{r_k}$ (the $k$-th column of $\underline{R}$) is an eigenvector corresponding to the $k$-th eigenvalue $\lambda_k$ (for $k = 1, \ldots, K$). Once obtaining the $\underline{R}$, the original conservative variable $\underline{U}$ can be projected into the space spanned by the $\underline{r}$'s, which gives the characteristic variable

$$\underline{V} := \underline{R}^{-1} \underline{U}, \tag{14}$$

The next step is then to apply the `ScalarWeno` limiter (Equations (6)–(10)) on each scalar component of $\underline{V}$, which gives $\underline{V}^{\mathrm{new}}$. After obtaining the reconstructed characteristic variable $\underline{V}^{\mathrm{new}}$, it can be turned back into the original conservative variable

$$\underline{U}^{\mathrm{new}}_{k \to i} := \underline{R}\,\underline{V}^{\mathrm{new}}, \tag{15}$$

in which the subscript $k \to i$ means that it is a function defined on $E_i$, which is constructed with the help of $E_k$. The final step is to weight these reconstructed conservative variables by the measure of the corresponding adjacent element:

$$\underline{U}^{\mathrm{new}}_i := \frac{\sum_{k \in K_i} \underline{U}^{\mathrm{new}}_{k \to i} |E_k|}{\sum_{k \in K_i} |E_k|}. \tag{16}$$

Since the eigenvalue decomposition (Equation (13)) plays a central role in this limiting procedure, we would like to name it as the `EigenWeno` limiter.

### 2.2.3. The `LazyWeno` Limiter

The `EigenWeno` limiter (Equations (11)–(16)) works well on two-dimensional meshes in [23,24] and on three-dimensional meshes in this article. However, it depends on the conservation law system to be solved and thus is not applicable if the eigenvalue decomposition (Equation (13)) is not easily computable, or the task is to design a limiter for general matrix-valued functions (not necessarily the conservative variable of a conservation law system). In either case, one could simply apply the `ScalarWeno` limiter (Equations (6)–(9)) to each scalar component of $\underline{U}$, which is a matrix-valued function. Since this limiting procedure requires less derivation and computational resources, we would like to name it as the `LazyWeno` limiter.

### 2.3. Temporal Discretization

Equation (4) is a typical nonlinear ordinary differential equation system, which can be solved by various numerical methods, such as the Runge–Kutta methods (see [33]). However, to preserve the total variation diminishing (TVD) property of the solution, the method itself should be TVD [34] and some kind of limiters (already discussed in Section 2.2) should be carefully incorporated into it. In this article, we follow the practice of [23,24], which use the explicit third-order TVD Runge–Kutta method:

$$
\begin{aligned}
\underline{\hat{U}}^{n+1/3} &= \underline{\hat{U}}^{n} + \underline{R}^{n}\Delta t \\
\underline{\hat{U}}^{n+2/3} &= \frac{3}{4}\underline{\hat{U}}^{n} + \frac{1}{4}\left(\underline{\hat{U}}^{n+1/3} + \underline{R}^{n+1/3}\Delta t\right) \\
\underline{\hat{U}}^{n+1} \equiv \underline{\hat{U}}^{n+3/3} &= \frac{1}{3}\underline{\hat{U}}^{n} + \frac{2}{3}\left(\underline{\hat{U}}^{n+2/3} + \underline{R}^{n+2/3}\Delta t\right)
\end{aligned}
\tag{17}
$$

in which, integers in superscripts are the marks of time steps, and fractions in superscripts represent intermediate stages. The values of the right hand side (RHS) expressions are not guaranteed to be TVD, so limiting procedures must be applied before assigning them the the left hand side (LHS). To make it more clear, we introduce a nonlinear operator $\mathcal{L}$ (stands for limiter) into the RHS of Equation (17), which gives

$$
\begin{aligned}
\underline{\hat{U}}^{n+1/3} &= \mathcal{L}\left[\underline{\hat{U}}^{n} + \underline{R}^{n}\Delta t\right] \\
\underline{\hat{U}}^{n+2/3} &= \mathcal{L}\left[\frac{3}{4}\underline{\hat{U}}^{n} + \frac{1}{4}\left(\underline{\hat{U}}^{n+1/3} + \underline{R}^{n+1/3}\Delta t\right)\right] \\
\underline{\hat{U}}^{n+1} \equiv \underline{\hat{U}}^{n+3/3} &= \mathcal{L}\left[\frac{1}{3}\underline{\hat{U}}^{n} + \frac{2}{3}\left(\underline{\hat{U}}^{n+2/3} + \underline{R}^{n+2/3}\Delta t\right)\right]
\end{aligned}
\tag{18}
$$

This notation clearly emphasize the application of limiters.

*2.4. Parallel Programming*

Both the flux integrals (Equation (5)) in the DG method and the function borrowing (Equation (6)) in WENO limiters put a requirement on each cell to access its neighbors in $\mathcal{O}(1)$ time, which is not supported by commonly used mesh formats. For this reason, we do not partition the input mesh directly using the `METIS_PartMeshDual(...)` function as in traditional finite element methods, but convert the mesh to its dual graph by the `METIS_MeshToDual(...)` function, and then partition the graph using the `METIS_PartGraphKway(...)` function. The `METIS_MeshToDual(...)` function stores the cell adjacency information in dynamically allocated arrays pointed by raw pointers, which should then be the released exactly once by some caller in the call stack. To avoid memory bugs, we suggest to wrap such raw pointers into some smart pointers, such as those provided by the standard library of modern C++ [35].

Once we obtain the partitioning, each part of the mesh should then be load by a process, which holds and updates local data sequentially and shares data on inter-part boundaries with neighbors when necessary. This is a typical scenario of the distributed memory parallelization, which achieves acceleration by solving relatively equal-sized subproblems simultaneously on multiple cores. Compared with this, the shared memory parallelization, which provides a global memory address space shared by multiple threads, is generally easier to program but less scalable. The price we paid for scalability is the explicit management of message passing for sharing data between processes. Thanks to the publicly available implementations of the MPI standard, such as MPICH (https://www.mpich.org) and Open-MPI (https://www.open-mpi.org), the code for doing this is much simpler than it used to be. To improve readability and maintainability of our code, we wrap these communication operations in functions names as `ShareSomething(...)`, which share the same code structure:

1. For each destination, put the data to be sent into a sending buffer and register a request of sending by calling the `MPI_Isend(...)` function.
2. For each source, allocate a receiving buffer for the data to be received and register a request of receiving by calling the `MPI_Irecv(...)` function.
3. Performed other computations that can be conducted without communications.
4. Block the process until all its requests complete by calling the the `MPI_Waitall(...)` function.

The third step is optional but may help to improve parallel efficiency, since it allows computations to overlap with communications.

## 3. Results

In this section, we give the results of various numerical experiments to show the accuracy and performance of the methods described in Section 2. Even though all these experiments can be carried out on one- or two-dimensional structured meshes, we intentionally solve them on three-dimensional unstructured meshes. In this way, the applicability of our solvers for real engineering problems could be demonstrated.

*3.1. Linear Conservation Laws*

The first group of problems to be solved is the linear version of Equation (1):

$$\partial_t \underline{U} + \underline{A}^x \partial_x \underline{U} + \underline{A}^y \partial_y \underline{U} + \underline{A}^z \partial_z \underline{U} = \underline{O} \tag{19}$$

with certain boundary and initial conditions. These problems are mathematically simple in the sense that they can be solved analytically. The existence of analytic solutions gives us a good way to measure the accuracy of our numerical solvers. In this subsection, we use tetrahedral meshes generated in a $[0,4] \times [0,1] \times [0,0.5]$ box.

3.1.1. Scalar Case

This is the simplest case of Equation (19):

**Problem 1.** *In Equation (19), let $\underline{U}$ consists only one component and each $\underline{A}$ consists a single number:*

$$\underline{U}(\vec{x}, t) = \left[U(x, y, z, t)\right], \qquad \underline{A}^x = \left[-10\right], \qquad \underline{A}^y = \underline{A}^z = \left[0\right].$$

*The following boundary conditions*

$$\underline{U}(x = 0, y, z, t) = \left[-10\right] =: \underline{U}_L, \qquad \underline{U}(x = 4, y, z, t) = \left[+10\right] =: \underline{U}_R,$$

*and the initial condition*

$$\underline{U}(x, y, z, t = 0) = \underline{U}_L$$

*are applied.*

The analytic solution of Problem 1 is:

$$\underline{U}(x, y, z, t) = \begin{cases} \underline{U}_L & x - 4 < -10t \\ \underline{U}_R & x - 4 > -10t \end{cases}$$

which can be interpreted as a left-running plane wave, whose profile is a jump initially positioned at the right end ($x = 4$). To compare the accuracy of various schemes, we plot the meshes and the numerical solutions at the same moment ($t = 0.2$) in Figures 1–4. The white-colored regions in these figures are continuous transition layers, which are inevitable due to the dissipation of numerical schemes. The thickness of such transition layer, however, can then be used as an indicator of the scheme's accuracy. Ideally, the thickness should be infinitesimal, as in the analytic solution. The differences between these results are more obvious in Figures 5 and 6, in which we plot the values on 1001 uniformly distributed sample points along the longitudinal axis (on which $y = 0.5$ and $z = 0.25$) for each solver.

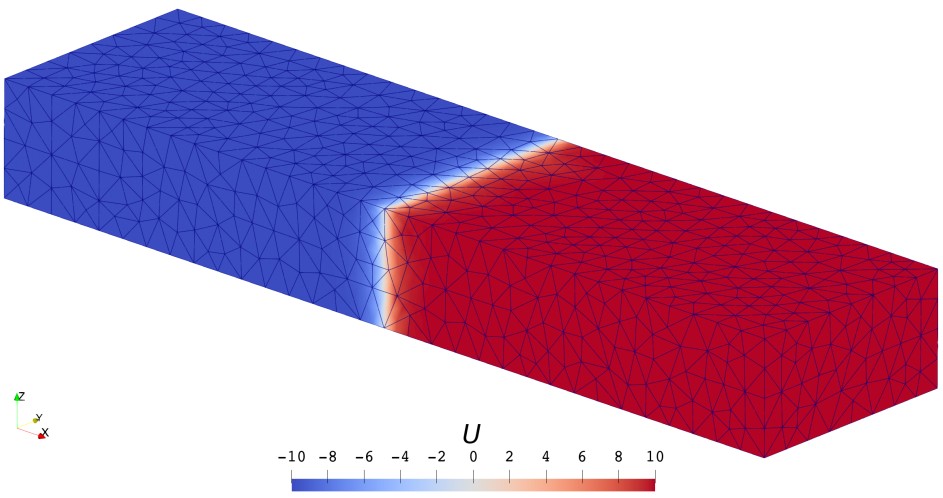

**Figure 1.** Third-order solution of Problem 1 on medium ($h \approx 2^{-3}$) cells.

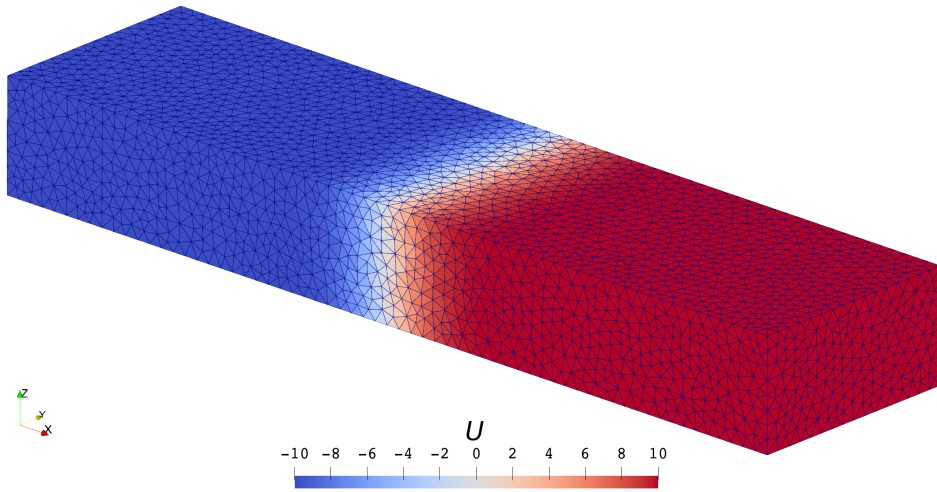

**Figure 2.** First-order solution of Problem 1 on small ($h \approx 2^{-4}$) cells.

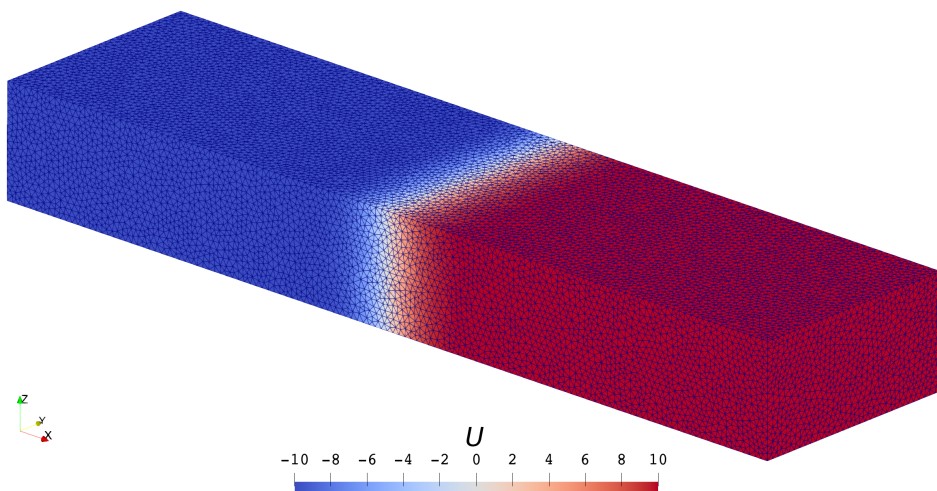

**Figure 3.** First-order solution of Problem 1 on tiny ($h \approx 2^{-5}$) cells.

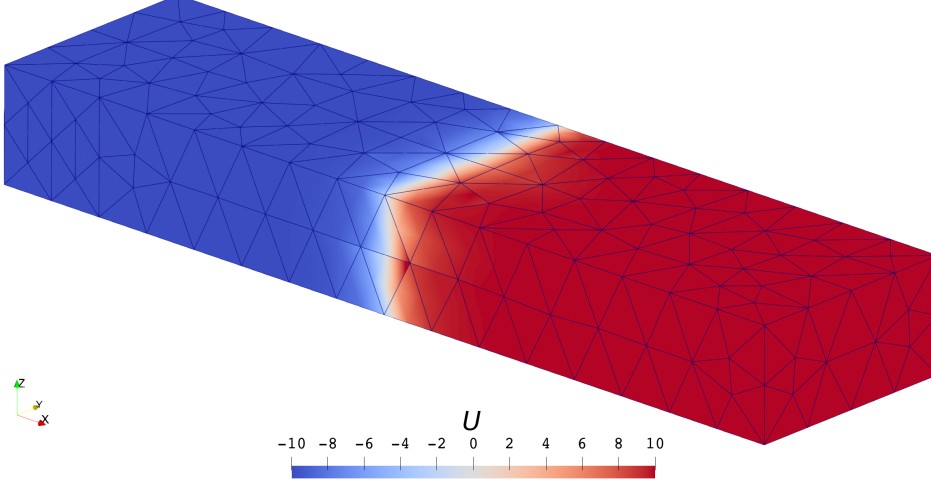

**Figure 4.** Third-order solution of Problem 1 on big ($h \approx 2^{-2}$) cells.

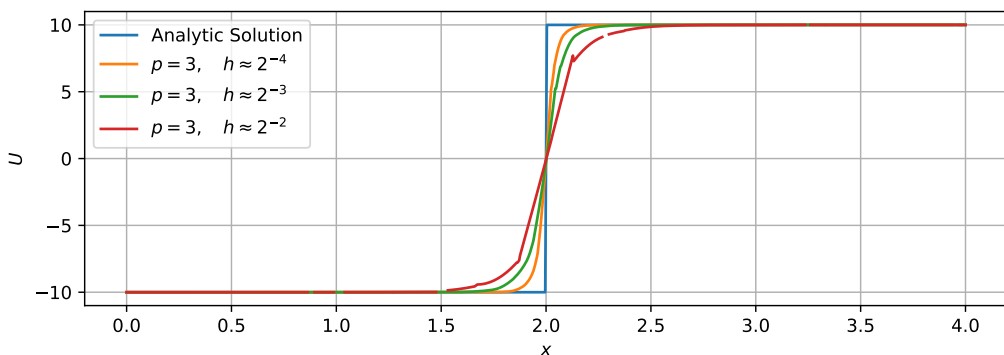

**Figure 5.** Comparison between solutions of Problem 1 given by running the same solver on different meshes.

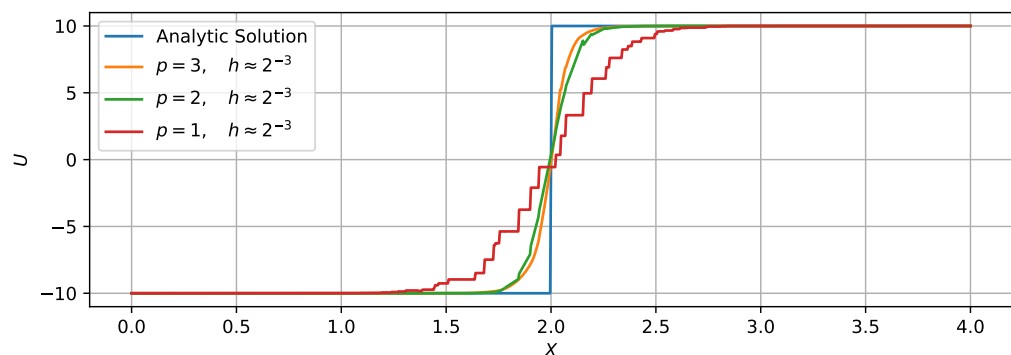

**Figure 6.** Comparison between solutions of Problem 1 given by running different solvers on the same mesh.

In Table 1, we show the measured error and time cost of each solver/mesh pair. The seconds consumed by first-order solutions are somewhat exaggerated, since we use the same high-order quadrature rules for both first- and third-order solvers, which is necessary to integrate non-constant errors. If we did not have to measure the errors, then low-order numerical integrators could be used, which might save some time.

**Table 1.** Accuracy and time cost of each solver–mesh ($p$–$h$) pair.

| $L_1$-Error with Respect to the Analytic Solution | | | | |
|---|---|---|---|---|
| $p$ ⟍ $h$ | $2^{-2}$ | $2^{-3}$ | $2^{-4}$ | $2^{-5}$ |
| 1 | 2.858 | 2.095 | 1.524 | 1.108 |
| 2 | 1.258 | 0.771 | 0.463 | 0.275 |
| 3 | 1.021 | 0.590 | 0.341 | |

| Time Cost (in Seconds) Measured on a Single Core Whose Main Frequency Is 2.7 GHz | | | | |
|---|---|---|---|---|
| $p$ ⟍ $h$ | $2^{-2}$ | $2^{-3}$ | $2^{-4}$ | $2^{-5}$ |
| 1 | 0.373 | 1.129 | 16.533 | 306.293 |
| 2 | 1.580 | 14.894 | 253.821 | 4986.391 |
| 3 | 4.147 | 61.425 | 906.914 | |

The following conclusions can be drawn from both Figures 1–6 and Table 1:

- Both mesh refinement (decreasing $h$) and order increment (increasing $p$) can help to improve accuracy.

- The solver of the highest order ($p = 3$) on the coarsest ($h \approx 2^{-2}$) mesh defeats the solver of the lowest order ($p = 1$) on the finest ($h \approx 2^{-5}$) mesh in accuracy but saves quite a lot of time.
- High-order schemes are better than low-order ones in the sense of getting the same level of accuracy with less time cost.

### 3.1.2. System Case

**Problem 2.** *In Equation (19), let $\underline{U}$ consists two components and each $\underline{A}$ be a $2 \times 2$ matrix:*

$$\underline{U}(\vec{x}, t) = \begin{bmatrix} U_1(x,y,z,t) \\ U_2(x,y,z,t) \end{bmatrix}, \qquad \underline{A}^x = \begin{bmatrix} 6 & -2 \\ -2 & 6 \end{bmatrix}, \qquad \underline{A}^y = \underline{A}^z = \begin{bmatrix} 0 & 0 \\ 0 & 0 \end{bmatrix},$$

*The following boundary conditions*

$$\underline{U}(x = 0, y, z, t) = \begin{bmatrix} 0 \\ 0 \end{bmatrix} =: \underline{U}_{\mathrm{L}}, \qquad \underline{U}(x = 4, y, z, t) = \begin{bmatrix} 12 \\ -4 \end{bmatrix} =: \underline{U}_{\mathrm{R}},$$

*and the initial condition*

$$\underline{U}(x, y, z, t = 0) = \underline{U}_{\mathrm{R}}$$

*are applied.*

To solve this problem analytically, we first obtain the eigenvalue decomposition of $\underline{A}^x$, which is

$$\underbrace{\begin{bmatrix} 6 & -2 \\ -2 & 6 \end{bmatrix}}_{\underline{A}^x} = \underbrace{\begin{bmatrix} 1 & 1 \\ -1 & 1 \end{bmatrix}}_{\underline{R}^x} \underbrace{\begin{bmatrix} 8 & \\ & 4 \end{bmatrix}}_{\underline{\Lambda}^x} \underbrace{\begin{bmatrix} 1/2 & -1/2 \\ 1/2 & 1/2 \end{bmatrix}}_{(\underline{R}^x)^{-1}}$$

By introducing the characteristic variable $\underline{V} := (\underline{R}^x)^{-1}\underline{U}$, which means

$$\begin{bmatrix} V_1 \\ V_2 \end{bmatrix} := \begin{bmatrix} 1/2 & -1/2 \\ 1/2 & 1/2 \end{bmatrix} \begin{bmatrix} U_1 \\ U_2 \end{bmatrix} = \frac{1}{2} \begin{bmatrix} U_1 - U_2 \\ U_1 + U_2 \end{bmatrix},$$

the boundary conditions become

$$\underline{V}_{\mathrm{L}} = (\underline{R}^x)^{-1}\underline{U}_{\mathrm{L}} = \begin{bmatrix} 0 \\ 0 \end{bmatrix}, \qquad \underline{V}_{\mathrm{R}} = (\underline{R}^x)^{-1}\underline{U}_{\mathrm{R}} = \begin{bmatrix} 8 \\ 4 \end{bmatrix},$$

and the system can be decoupled:

$$\begin{bmatrix} \partial_t + 8\partial_x & \\ & \partial_t + 4\partial_x \end{bmatrix} \begin{bmatrix} V_1 \\ V_2 \end{bmatrix} = \begin{bmatrix} 0 \\ 0 \end{bmatrix}.$$

We can then solve these two scalar problems independently, which gives

$$V_1(x, y, z, t) = \begin{cases} 0 & x < 8t \\ 8 & x > 8t \end{cases}, \qquad V_2(x, y, z, t) = \begin{cases} 0 & x < 4t \\ 4 & x > 4t \end{cases}.$$

The solution of Problem 2 can be obtained by $\underline{U} = \underline{R}^x \underline{V}$, which gives

$$\underline{U}(\vec{x}, t) = \begin{bmatrix} V_2 + V_1 \\ V_2 - V_1 \end{bmatrix} = \begin{cases} \underline{U}_{\mathrm{L}} & x/t < 4 \\ \underline{U}_{\mathrm{M}} & x/t \in (4, 8), \\ \underline{U}_{\mathrm{R}} & x/t > 8 \end{cases} \tag{20}$$

where

$$\underline{U}_{\mathrm{L}} = \begin{bmatrix} 0 \\ 0 \end{bmatrix}, \qquad \underline{U}_{\mathrm{M}} = \begin{bmatrix} 4 \\ 4 \end{bmatrix}, \qquad \underline{U}_{\mathrm{R}} = \begin{bmatrix} 12 \\ -4 \end{bmatrix}.$$

With this analytic solution, we can evaluate the accuracy of our numerical solvers. Four solver–limiter pairs are tested on the same mesh ($h \approx 2^{-3}$) used in Figure 1.

We plot the contour of $\underline{U}(x, y, z, t = 0.3)$ with the underlying mesh in Figures 7 and 8 and compare the results along the longitudinal axis at $t = 0.3$ with the analytic solution in Figures 9 and 10. It is clear that both `LazyWeno` and `EigenWeno` (see Section 2.2) can essentially suppress non-physical oscillations in each component. Figure 11 shows that higher-order ($p = 3$) solvers still outperforms lower-order ($p = 2$) solvers in accuracy and the `EigenWeno` limiter generally works better than its `LazyWeno` counterpart. For this reason, we will use `EigenWeno` limiters exclusively in the rest of this section.

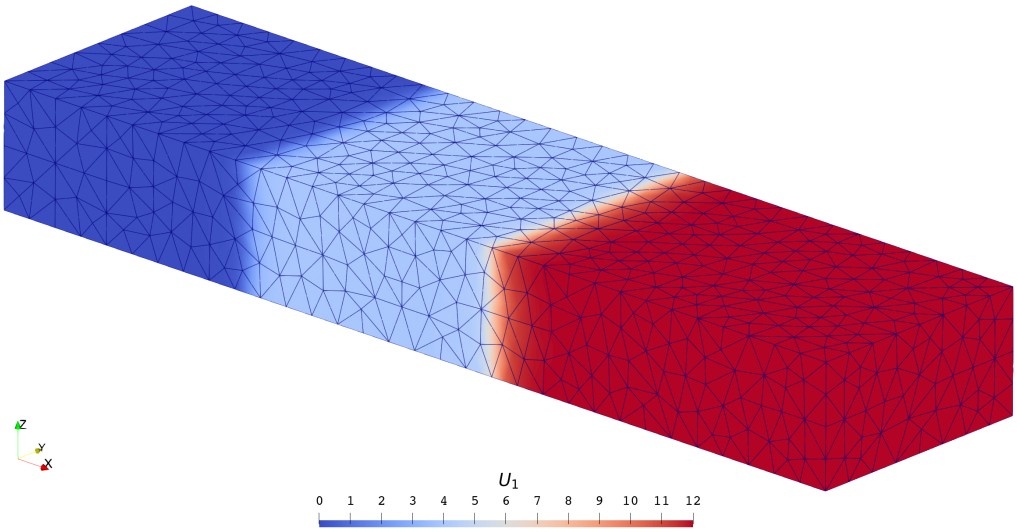

**Figure 7.** Third-order solution of $U_1(t = 0.3)$ in Problem 2.

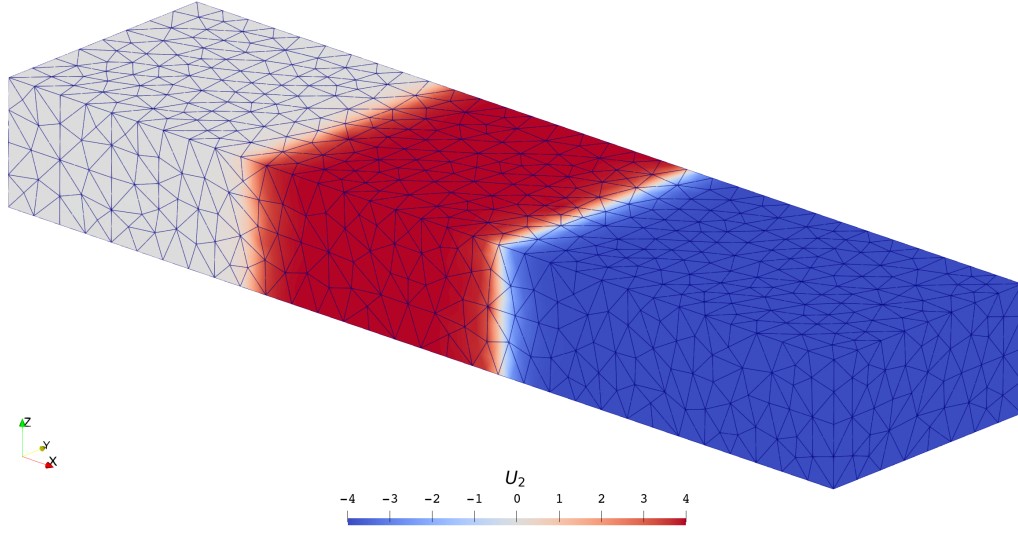

**Figure 8.** Third-order solution of $U_2(t = 0.3)$ in Problem 2.

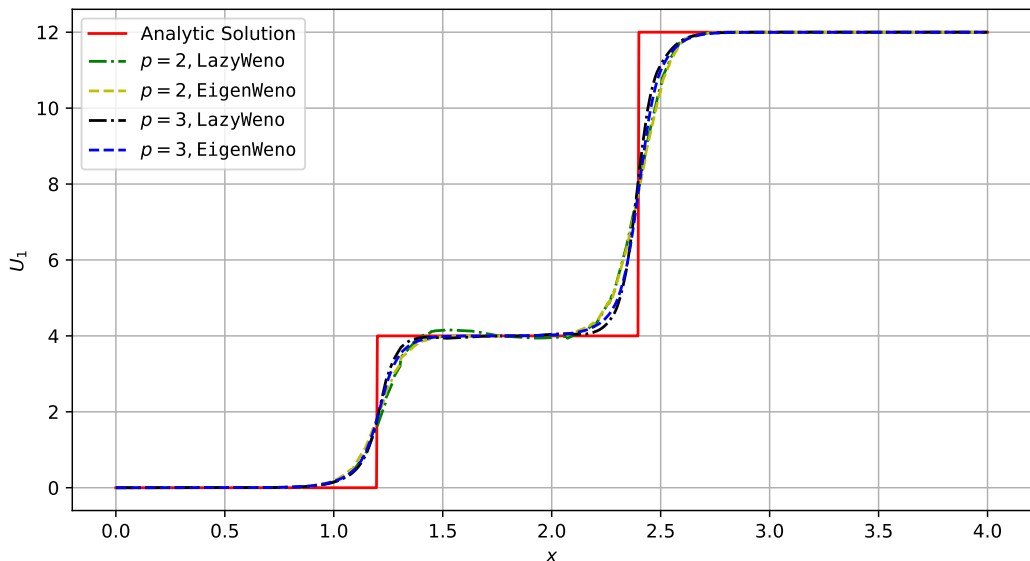

**Figure 9.** Comparison between solutions of $U_1(t = 0.3)$ in Problem 2.

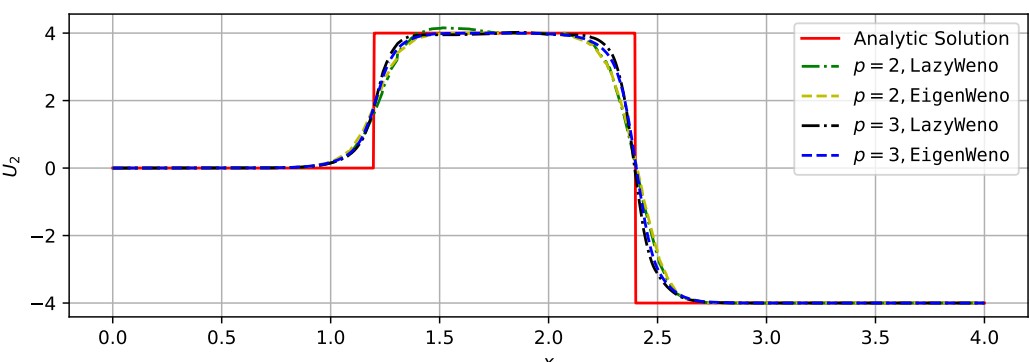

**Figure 10.** Comparison between solutions of $U_2(t = 0.3)$ in Problem 2.

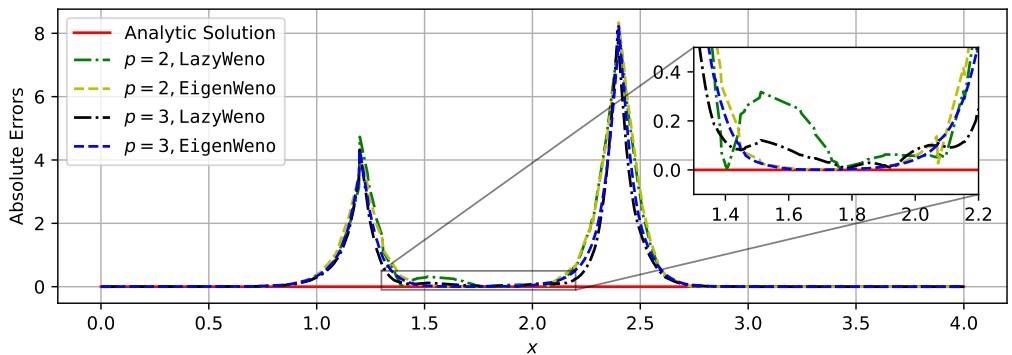

**Figure 11.** Comparison between absolute errors of numerical solutions in Figures 9 and 10.

### 3.2. Inviscid Compressible Flows

The second group of problems to be solved is the three-dimensional Euler system:

$$\partial_t \begin{bmatrix} \rho \\ \rho u_x \\ \rho u_y \\ \rho u_z \\ \rho e_0 \end{bmatrix} + \partial_x \begin{bmatrix} \rho u_x \\ \rho u_x u_x + p \\ \rho u_y u_x \\ \rho u_z u_x \\ \rho h_0 u_x \end{bmatrix} + \partial_y \begin{bmatrix} \rho u_y \\ \rho u_x u_y \\ \rho u_y u_y + p \\ \rho u_z u_y \\ \rho h_0 u_y \end{bmatrix} + \partial_z \begin{bmatrix} \rho u_z \\ \rho u_x u_z \\ \rho u_y u_z \\ \rho u_z u_z + p \\ \rho h_0 u_z \end{bmatrix} = \begin{bmatrix} 0 \\ 0 \\ 0 \\ 0 \\ 0 \end{bmatrix} \qquad (21)$$

with certain boundary and initial conditions. These problems are genuinely nonlinear which cannot be solved analytically in general. However, their exact or high-order solutions in lower-dimensional spaces are well known in CFD studies, which can still be used to test our three-dimensional solvers.

For this system, the $\underline{A}^\nu$ defined in Equation (12) depends on $\underline{U}$, so do the $\underline{R}^{-1}$ in Equation (14) and the $\underline{R}$ in Equation (15). Fortunately, these matrices can be explicitly formulated:

$$
\underline{R}(\underline{U}) = \begin{bmatrix} 1 & 1 & 0 & 0 & 1 \\ u_x - av_x & u_x & \sigma_x & \pi_x & u_x + av_x \\ u_y - av_y & u_y & \sigma_y & \pi_y & u_y + av_y \\ u_z - av_z & u_z & \sigma_z & \pi_z & u_z + av_z \\ h_0 - u_\nu a & \frac{u_x^2 + u_y^2 + u_z^2}{2} & u_\sigma & u_\pi & h_0 + u_\nu a \end{bmatrix}, \qquad \begin{bmatrix} u_\nu \\ u_\sigma \\ u_\pi \end{bmatrix} = \begin{bmatrix} v_x & \sigma_x & \pi_x \\ v_y & \sigma_y & \pi_y \\ v_z & \sigma_z & \pi_z \end{bmatrix}^{-1} \begin{bmatrix} u_x \\ u_y \\ u_z \end{bmatrix},
$$

$$
\underline{L}(\underline{U}) = \begin{bmatrix} \frac{1}{2}\left(B_2 + \frac{u_\nu}{a}\right) & \frac{-1}{2}\left(B_1 u_x + \frac{v_x}{a}\right) & \frac{-1}{2}\left(B_1 u_y + \frac{v_y}{a}\right) & \frac{-1}{2}\left(B_1 u_z + \frac{v_z}{a}\right) & \frac{1}{2}B_1 \\ 1 - B_2 & B_1 u_x & B_1 u_y & B_1 u_z & -B_1 \\ -u_\sigma & \sigma_x & \sigma_y & \sigma_z & 0 \\ -u_\pi & \pi_x & \pi_y & \pi_z & 0 \\ \frac{1}{2}\left(B_2 - \frac{u_\nu}{a}\right) & \frac{-1}{2}\left(B_1 u_x - \frac{v_x}{a}\right) & \frac{-1}{2}\left(B_1 u_y - \frac{v_y}{a}\right) & \frac{-1}{2}\left(B_1 u_z - \frac{v_z}{a}\right) & \frac{1}{2}B_1 \end{bmatrix},
$$

in which $B_1 := (\gamma - 1)/a^2$ and $B_2 := B_1(u_\nu^2 + u_\sigma^2 + u_\pi^2)$.

3.2.1. Shock Tube Problems

These problems are usually defined as one-dimensional problems, but we treat them as three-dimensional ones. All these problems are considered in a $[0.0, 5.0] \times [0.0, 1.0] \times [0.0, 0.5]$ box with all boundaries closed but the left and right ends open. Although no analytic solutions exist, we can still use the method described in [31], which solves nonlinear algebraic equations numerically, to obtain their exact solutions. To test the numerical methods described in Section 2, we use the unstructured hexahedral mesh in Figure 12, in which $h \approx 1/10$.

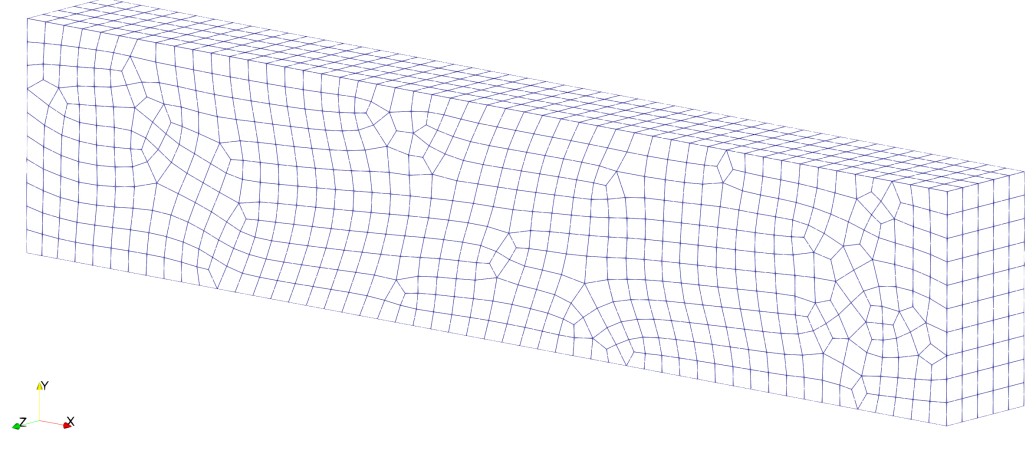

**Figure 12.** Mesh for Problems 3–5.

**Problem 3** (Sod). *Solve the Euler system (Equation (21)) for $t \in [0.0, 1.0]$ with the initial condition*

$$
\begin{bmatrix} \rho & u & vs. & w & p \end{bmatrix}_{t=0} = \begin{cases} \begin{bmatrix} 1.000 & 0.000 & 0.000 & 0.000 & 1.000 \end{bmatrix} & x < 2 \\ \begin{bmatrix} 0.125 & 0.000 & 0.000 & 0.000 & 0.100 \end{bmatrix} & x > 2 \end{cases}
$$

**Problem 4** (Lax). *Solve the Euler system (Equation (21)) for $t \in [0.0, 0.6]$ with the initial condition*

$$
\begin{bmatrix} \rho & u & vs. & w & p \end{bmatrix}_{t=0} = \begin{cases} \begin{bmatrix} 0.445 & 0.698 & 0.000 & 0.000 & 3.528 \end{bmatrix} & x < 2 \\ \begin{bmatrix} 0.500 & 0.000 & 0.000 & 0.000 & 0.571 \end{bmatrix} & x > 2 \end{cases}
$$

**Problem 5** (Vacuum). *Solve the Euler system (Equation (21)) for $t \in [0.0, 0.3]$ with the initial condition*

$$
\begin{bmatrix} \rho & u & vs. & w & p \end{bmatrix}_{t=0} = \begin{cases} \begin{bmatrix} 1.0 & -4.0 & 0.0 & 0.0 & 0.4 \end{bmatrix} & x < 2 \\ \begin{bmatrix} 1.0 & +4.0 & 0.0 & 0.0 & 0.4 \end{bmatrix} & x > 2 \end{cases}
$$

In Figures 13–15, we plot the density contours given by the same third-order solver with an `EigenWeno` limiter. In Figures 13–18, we plot the density distributions along the longitudinal axis (on which $y = 0.5$ and $z = 0.25$) of the box. All these results show that higher-order ($p = 3$) solvers with `EigenWeno` limiters are better than lower-order ($p = 1$) solvers at capturing discontinuities (shocks, contacts, expansions), which may occur frequently in compressible flows.

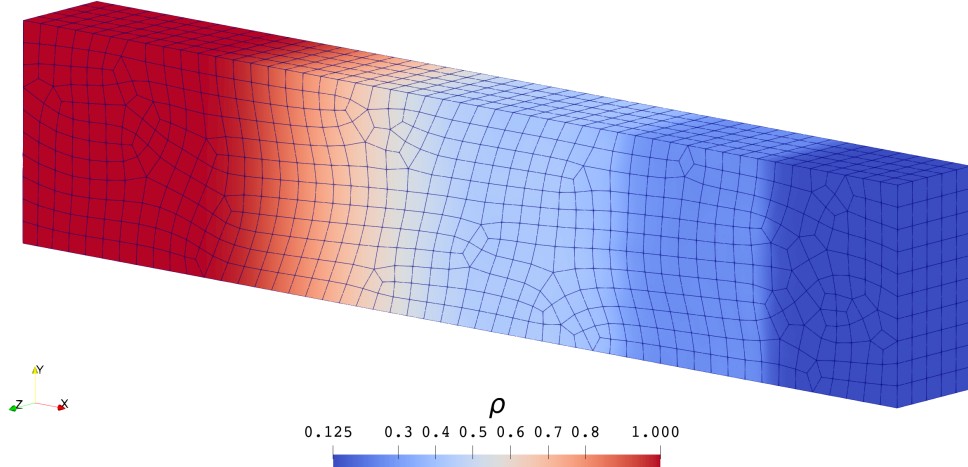

**Figure 13.** Third-order solution of $\rho(t = 1.0)$ in Problem 3.

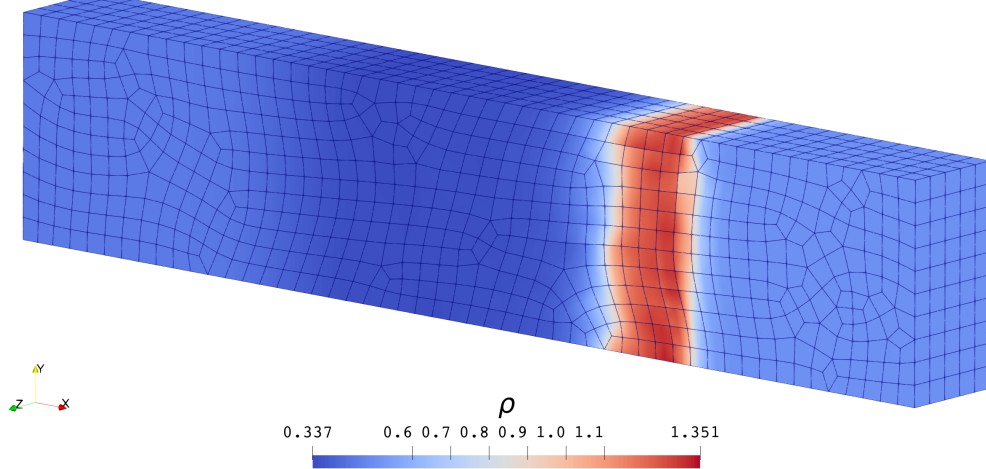

**Figure 14.** Third-order solution of $\rho(t = 0.6)$ in Problem 4.

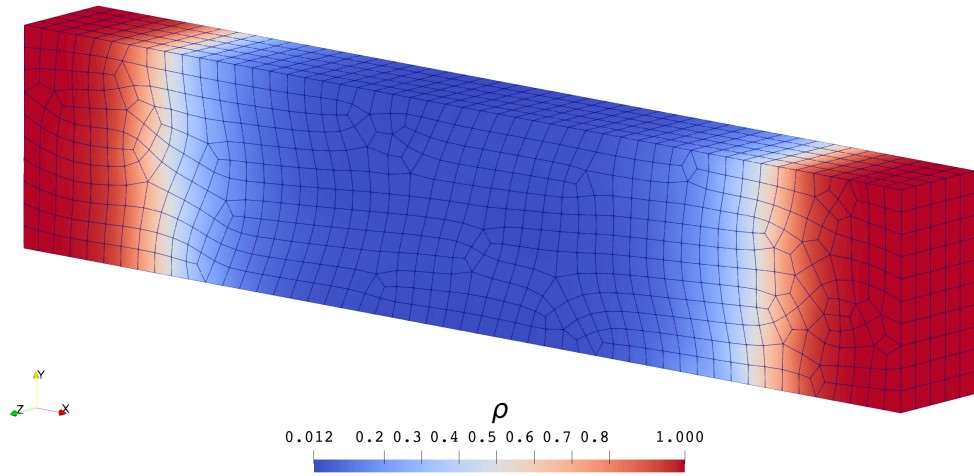

**Figure 15.** Third-order solution of $\rho(t = 0.3)$ in Problem 5.

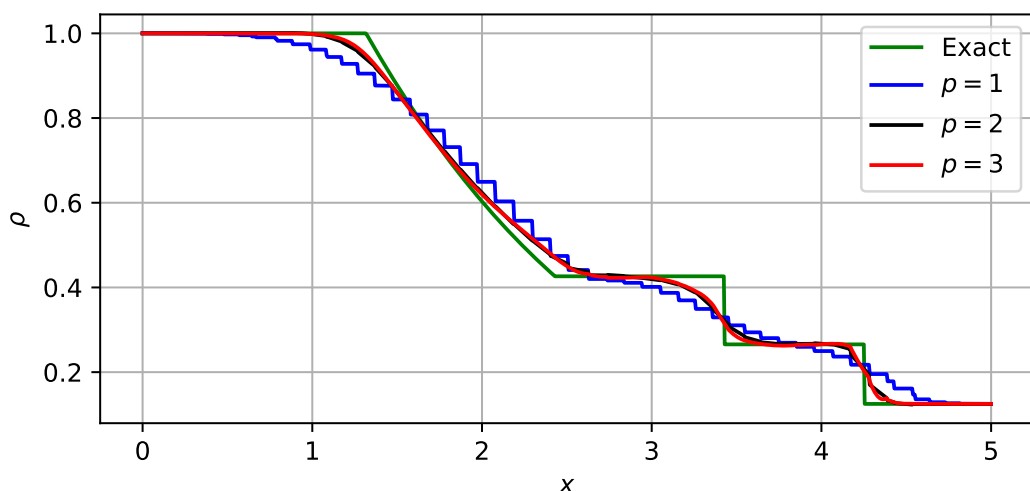

**Figure 16.** Comparison between solutions of $\rho(t = 1.0)$ in Problem 3.

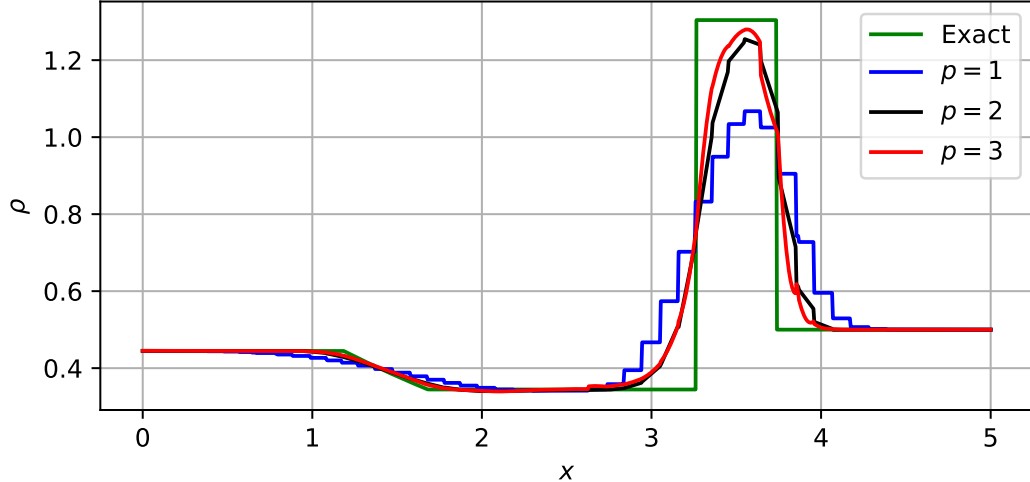

**Figure 17.** Comparison between solutions of $\rho(t = 0.5)$ in Problem 4.

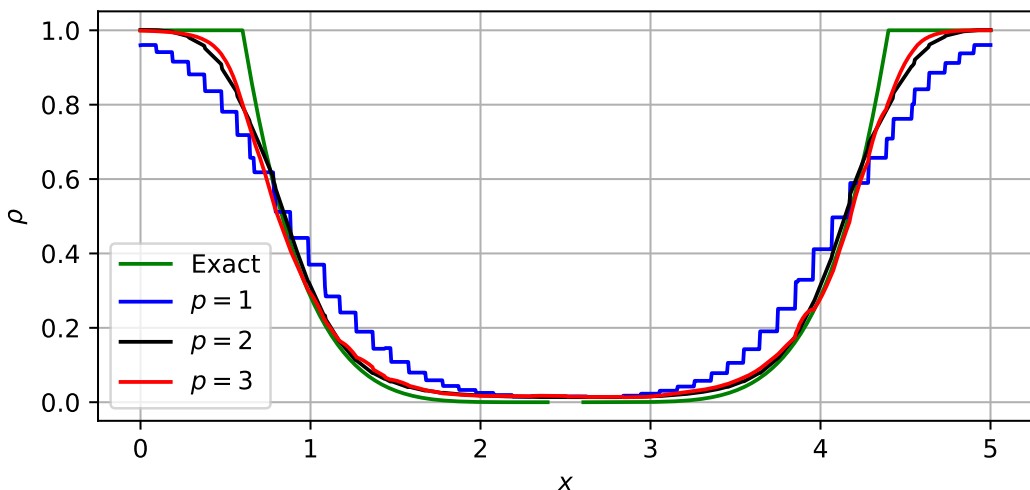

**Figure 18.** Comparison between solutions of $\rho(t = 0.3)$ in Problem 5.

### 3.2.2. Double Mach Reflection Problem

This is a classical two-dimensional problem originally proposed in [36], which we redefine here as a three-dimensional one:

**Problem 6.** *Solve the Euler system (Equation (21)) in the region defined in Figure 19, in which $x_0 = 1/6$. The initial condition is given as a moving shock wave:*

$$
\begin{bmatrix} \rho & u & vs. & w & p \end{bmatrix}_{t=0} = \begin{cases} \begin{bmatrix} 1.4 & 0.0 & 0.0 & 0.0 & 1.0 \end{bmatrix} & y < \sqrt{3}(x - x_0) \\ \begin{bmatrix} 8.0 & u_A & v_A & 0.0 & 116.5 \end{bmatrix} & y > \sqrt{3}(x - x_0) \end{cases} \tag{22}
$$

*in which $u_A = 4.125\sqrt{3}$ and $v_A = -4.125$ are the velocity components after the shock wave. The boundary conditions are given as following:*

- *The $x = 0$ surface is open as an inlet;*
- *The $x = 4$ surface and the $x < x_0$ part of the $y = 0$ surface are open as outlets;*
- *The $x > x_0$ part of the $y = 0$ surface is closed as a solid wall;*
- *The $y = 1$ surface has the following prescribed state:*

$$
\begin{bmatrix} \rho & u & vs. & w & p \end{bmatrix} = \begin{cases} \begin{bmatrix} 1.4 & 0.0 & 0.0 & 0.0 & 1.0 \end{bmatrix} & 1 < \sqrt{3}(x - (x_0 + u_A t)) \\ \begin{bmatrix} 8.0 & u_A & v_A & 0.0 & 116.5 \end{bmatrix} & 1 > \sqrt{3}(x - (x_0 + u_A t)) \end{cases}
$$

*which is consistent with the initial condition (Equation (22)).*

As a common practice, we plot the density contour at $t = 0.2$ in a $[0, 3] \times [0, 1]$ rectangle (on the $z = 0$ surface) for each solver in Figures 20–22. It is clear that as the accuracy order increases, the thickness of each discontinuity decreases and the rolled-up vortex structure becomes more clear.

Before concluding this section, we provide the measured performance of our third-order solver that produces Figure 22 in Table 2, in which

- $P$ means the number of processes (one process per core).
- $T_n$ means the wall clock time to finish the first $n$ step.
- $P \frac{T_{m+m} - T_n}{m}$ is the core time per step. The total core time of all steps is often used as an index for charging by high performance computing centers.

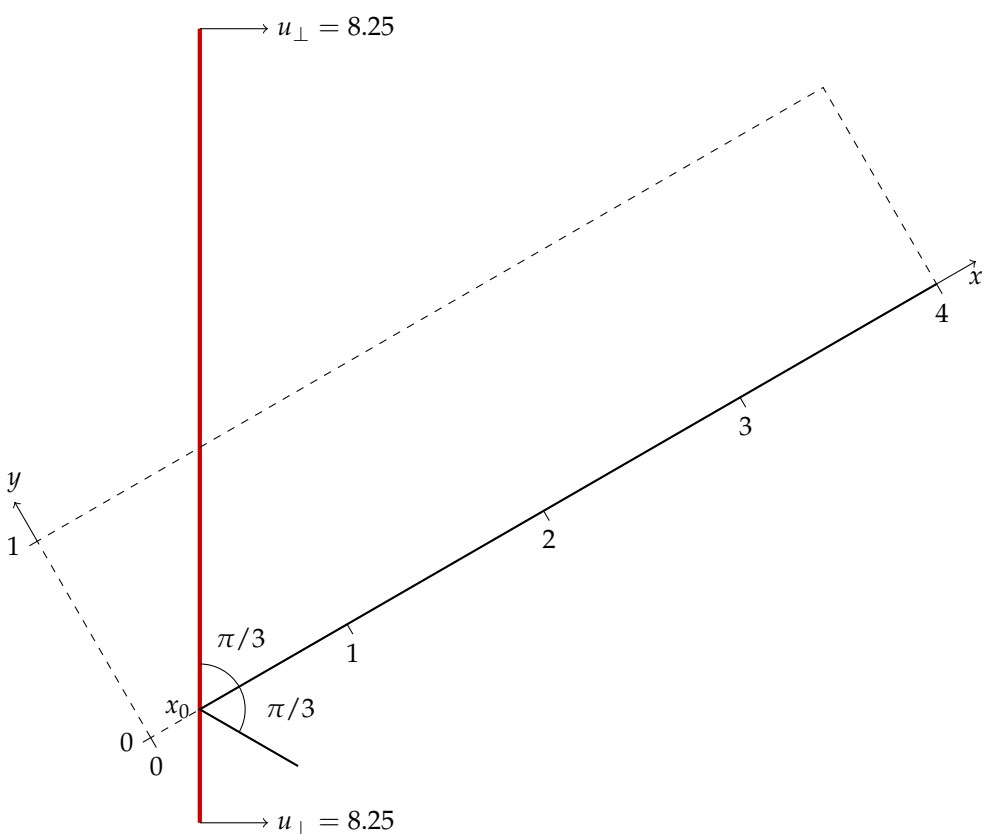

**Figure 19.** A schematic diagram of Problem 6. The rectangle bounded by four dashed lines and a solid line is the computational domain. The thick red line represents the initial shock wave, which is at an angle of $\pi/3$ relative to the $x$-axis.

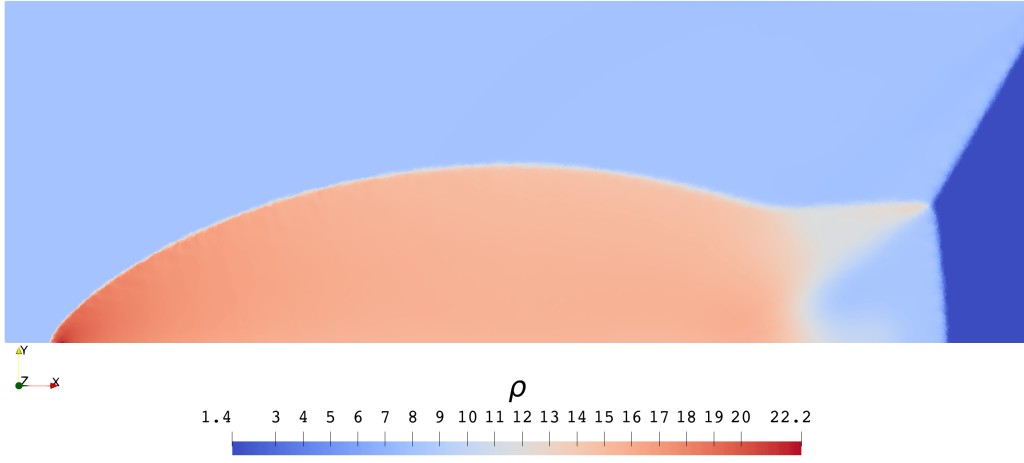

**Figure 20.** First-order solution of $\rho(x, y, z = 0, t = 0.2)$ in Problem 6 ($h \approx 1/200$).

Since parallel I/O operations require many collective communications, we write one frame every 100 steps. Thus, the difference between the values in the last two columns is the amortized core time of writing per step. We have to admit that this cost is growing as the number of cores increases. If the number of cores keeps increasing, this may be a bottleneck of maintaining scalability.

The community of parallel computing usually use the speedup ($S$) and efficiency ($E$) defined as

$$S = \frac{T_{\text{serial}}}{T_{\text{parallel}}}, \qquad E = \frac{S}{P} \times 100\%,$$

to assess the performance a parallel program. We follow this practice, calculate these values based on the measured data given in Table 2 and plot them in Figures 23 and 24. These figures show again that the I/O operations have adverse effects on the parallel performance.

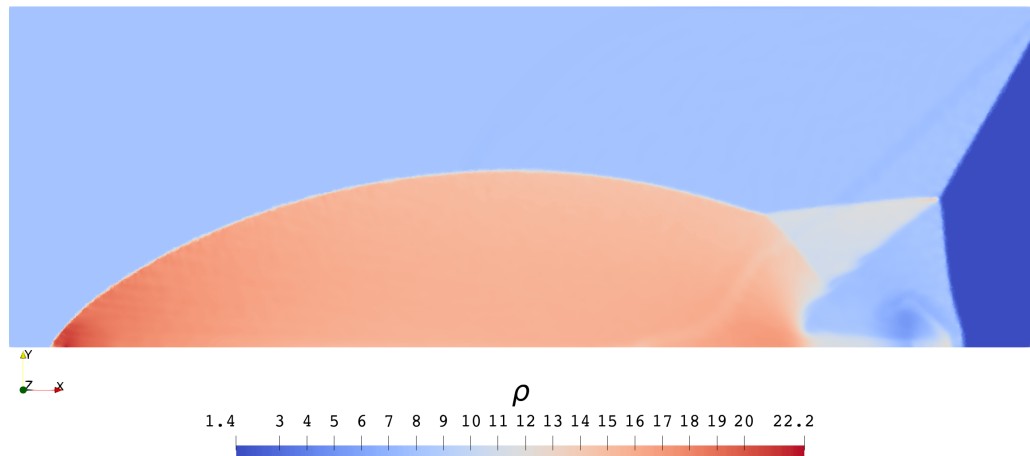

**Figure 21.** Second-order solution of $\rho(x, y, z = 0, t = 0.2)$ in Problem 6 ($h \approx 1/200$).

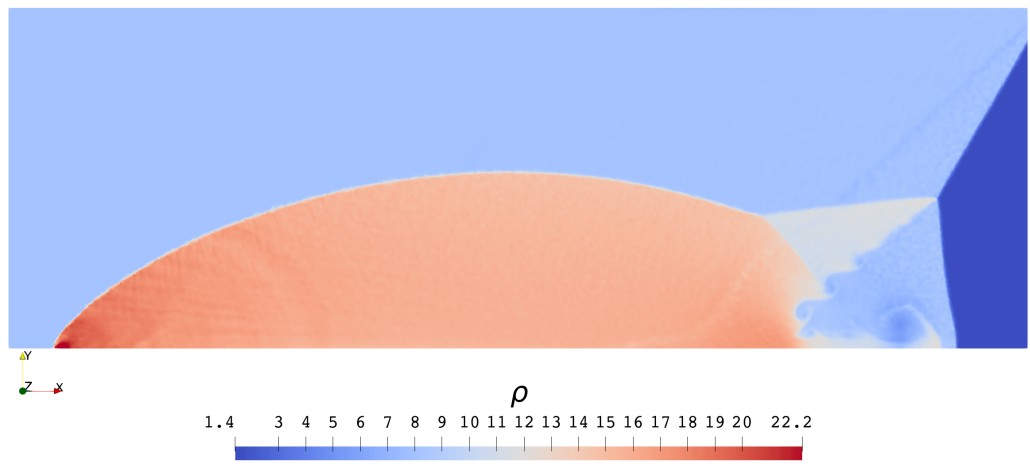

**Figure 22.** Third-order solution of $\rho(x, y, z = 0, t = 0.2)$ in Problem 6 ($h \approx 1/200$).

**Table 2.** Performance of the same solver running on different number of cores.

| $P$ | $T_{100}$ | $T_{199}$ | $T_{200}$ | $P\frac{T_{199}-T_{100}}{99}$ | $P\frac{T_{200}-T_{100}}{100}$ |
|---|---|---|---|---|---|
| 1 | 17,652.2 | 35,324.5 | 35,519.3 | 178.508 | 178.671 |
| 20 | 960.443 | 1912.365 | 1926.584 | 192.307 | 193.228 |
| 40 | 491.070 | 971.710 | 983.933 | 194.198 | 197.145 |
| 60 | 335.651 | 666.548 | 676.930 | 200.544 | 204.767 |
| 80 | 251.548 | 494.541 | 504.951 | 196.358 | 202.722 |
| 100 | 202.789 | 397.641 | 408.126 | 196.820 | 205.337 |

The efficiency values given in Figure 24 fluctuate around 90%, which are not as good as those above 99% in [26]. One source of such gap is the imperfect load balancing of our tests. Since we are using a three-dimensional unstructured mesh, it can hardly be partitioned uniformly, which is an *NP*-hard problem. On the other hand, their meshes are one- and two-dimensional structured, on which uniform partitioning can be trivially achieved. Figure 25 gives the distribution of cells of the 100-part mesh partitioning (Figure 26) used in this section, which shows a 3% fluctuation.

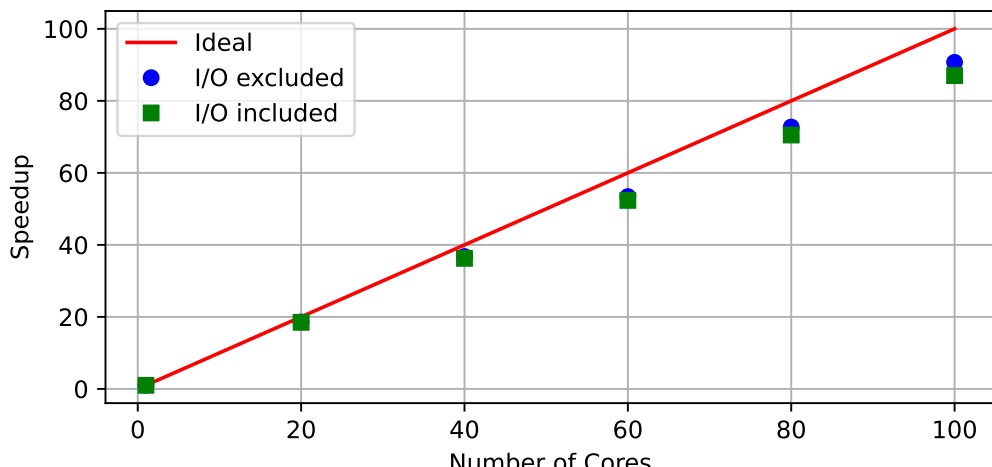

**Figure 23.** Speedup of the third-order solver for generating Figure 22.

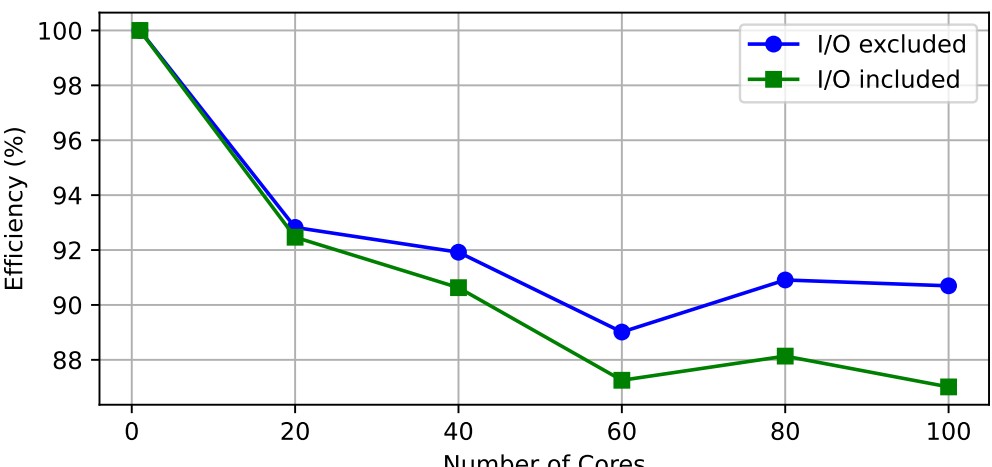

**Figure 24.** Parallel efficiency of the third-order solver for generating Figure 22.

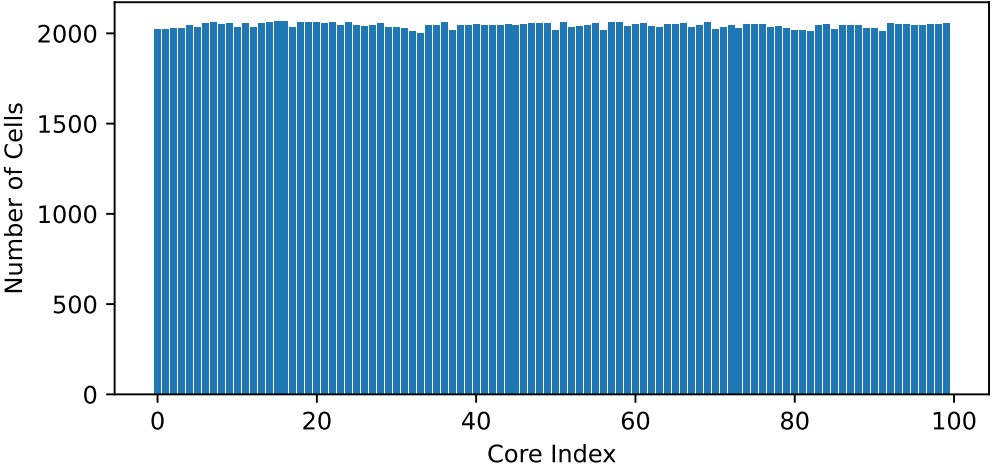

**Figure 25.** Distribution of cells in the mesh partitioning given in Figure 26.

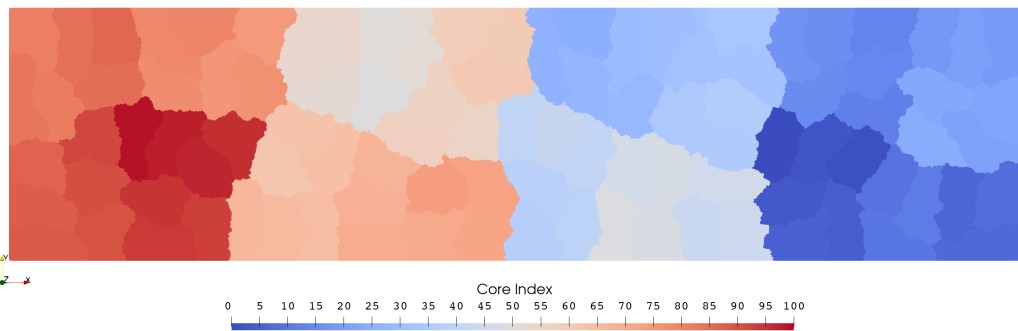

**Figure 26.** A 100-part mesh partitioning of the unstructured mesh used for generating Figures 20–22.

## 4. Discussion

In this article, we have formulated the RKDG methods and the WENO limiters for three-dimensional unstructured meshes. The algorithms have been implemented on top of the MPI standard, which supports distributed memory parallelization. The numerical experiments have shown that increasing a solver's accuracy order helps more to produce better results than just refining the mesh it uses. The efficient parallel implementation has made the time cost affordable for large problems, as long as the solvers can be executed on sufficiently large number of cores. Extending the methods to Navier–Stokes equations and applications of these high-order parallel solvers to real engineering problems are ongoing works. Further optimization of the parallel I/O module may be conducted to achieve better parallel performance.

**Author Contributions:** Conceptualization, W.P. and S.L.; methodology, W.P.; software, W.P. and Y.J.; validation, W.P. and Y.J.; formal analysis, W.P. and Y.J.; investigation, W.P.; resources, W.P.; data curation, W.P.; writing—original draft preparation, W.P.; writing—review and editing, W.P., Y.J. and S.L.; visualization, W.P.; supervision, S.L.; project administration, S.L.; funding acquisition, S.L. All authors have read and agreed to the published version of the manuscript.

**Funding:** This research was funded by the National High-tech R&D Program of China (863 Program) grant number 2012AA112201.

**Institutional Review Board Statement:** Not applicable.

**Informed Consent Statement:** Not applicable.

**Data Availability Statement:** The data presented in this article are all generated from the source code publicly available in our Git repository https://github.com/pvc1989/miniCFD accessed on 15 March 2022 (or the mirror site https://gitee.com/pvc1989/miniCFD, accessed on 15 March 2022).

**Acknowledgments:** The authors would like to express the deepest appreciation to Zhou Yukai for his professional assistance with typesetting and graphing.

**Conflicts of Interest:** The authors declare no conflict of interest. The funders had no role in the design of the study; in the collection, analyses, or interpretation of data; in the writing of the manuscript; or in the decision to publish the results.

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
