# Peer review of "An Efficient Parallel Implementation of the Runge–Kutta Discontinuous Galerkin Method with Weighted Essentially Non-Oscillatory Limiters on Three-Dimensional Unstructured Meshes"

_applsci, doi:10.3390/app12094228_

Round 1
Reviewer 1 Report
Review Summary
Title: “An Efficient Parallel Implementation of the Runge–Kutta Discontinuous Galerkin Method with Weighted Essentially Non-oscillatory Limiters on Three-dimensional Unstructured Meshes”
Authors: Weicheng Pei, Yuyan Jiang, and Shu Li
Summary: The paper describes the formulation and parallel implementation of discontinuous Galerkin (DG) method for unstructured meshes. In this manuscript, the formulation and the implementation are described in sufficient details, and results are presented for testcases which has analytical solutions. The results from the testcases are very promising. However, I would like the authors to address the following comments. I recommend to publish the paper after addressing the comments.
Comments:
1) There are many versions of DG methods available in the literature, with varying degrees of success. There is not enough information given in the introduction section on salient published papers on DG methods. I recommend the authors to conduct a thorough literature search on DG methods and summarize the findings.
2) On line numbers 115 and 116, it was stated that the EigenWeno template is taken from references 10 and 11. Since there are multiple graphs compare EigenWeno and LazyWeno, it is better to include a couple of sentences on the difference between these implementations in the manuscript.
3) Figure 5 compares the accuracy of simulation for different values of the order of the scheme and the mesh size. It will be easier to see the effect of these variables on accuracy of simulation by splitting this figure into two. In one figure keep h constant and vary p. In the second one, keep p constant and vary h. Same with Table 1.
4) For the mesh used for the shock tube problem, the mesh is extruded in z-direction with two layers of element. In this case, all elements have at least one boundary face. Therefore, any inconsistency in the boundary condition will be reflected in the solution used for comparison with the analytical results. Therefore, it is better to have a minimum of three layers of elements in the extruded direction, and take the data from the middle layer for comparison with analytical data.
Author Response
Thanks for the comments.
Response 1) Thanks for the suggestion. The introduction section has been revised to include more versions of DG methods and limiters. Although it may still looks non-exhaustive, this version has covered the most important papers in the development of the RKDG methods.
Response 2) Sorry for the misleading description in the original manuscript. Both the EigenWeno and LazyWeno templates are implemented by ourselves. The references only give the formulation of the 2d EigenWeno limiter. We don't know the details of their implementation. In the revised manuscript, we first give the unified formulation of the EigenWeno limiter for both 2d and 3d problems, and then explain the naming, and propose the LazyWeno limiter in a new subsection.
Response 3) Figure 5 has been split into two figures and Table 1 has been split into two (sub)tables. More solver-mesh pairs have been included, so that the effect of changing h and p can be seen more easily.
Response 4) The simulations have been reconducted on a new mesh and the corresponding figures have been updated. The new mesh has five layers in the extruded direction, so that about 3/5 of the elements have no boundary face.
Reviewer 2 Report
The method is not original but has the merit to do 3d in parallel
(note however that 3D, in particular with real 3d applciaitons exist in the litterature)
EigenWeno and Lazyweno methods are not suffisantly described
some typoes (check the full paper, please)
l38 applitions
l121 gauranteed
l212 Fgure
Author Response
Thanks for the comments.- This paper is mainly on method implementation and code validation. The application to real 3d problems is ongoing and will be published soon in the near future.
- In the revised manuscript, the section of limiting procedures has been split into three subsections. We first describe the limiter for scalar-valued functions and name it as the ScalarWeno limiter. In the second subsection, we give the formulation of the EigenWeno limiter, which is based on the eigenvalue decomposition of the flux Jacobian. The last subsection is for the LazyWeno limiter, in which the ScalarWeno limiter is applied directly to each scalar component of a matrix-valued function.
- The revised manuscript has been checked carefully.
Reviewer 3 Report
This study aims to incorporate unstructured mesh partitioning and message passing into the algorithms and implement them on top of publicly available libraries to support parallel execution.
The paper topic has been appropriate and worthy to be discussed in computing science. The results of various numerical experiments of the methods are well described. However, the previous research gap in parallel computing needs to be explained. The computation validation and data verification in this manuscript version are limited. The discussion and analysis need to be more profound.
Therefore, the recommendation for the publication is a major revision, and provide the following comments for correction.
- I would strongly advise the author to rewrite their introduction by identifying the previous research gap in parallel computing. Moreover, the paper's novelty and benefit could be well stated.
- The paper uses a computational simulation to analyze the result. Therefore, the numerical model should be explained in detail, including the algorithm design. Did the authors do the conservation analysis, which depicts that the inlet value and outlet section are the same? Is the numerical solution considered an experiment since the author stated "numerical experiment"?
- The validation of the simulation has been described. Moreover, the limitation of the data results and the simulation should be stated in the paper. I strongly suggested adding a discussion between nonparallel and parallel computing time. The aim is to accelerate calculation to produce an affordable cost; then, while it needs more cores, does it increases the cost while reducing the efficiency?. Does the best way is to use the same number of cores but accelerate the calculation; it makes sense that increasing cost will reduce the numerical duration. Did the authors do the numerical stability following the number increase of order and cores?
- This result and discussion should be extended by linking the results with the previous research and needing a deeper analysis.
Author Response
Thanks for the comments.- Thanks for the suggestion. We have included more references on parallel implementation of DG methods and summarized their merits and drawbacks.
- In the revised version, we have given more details of the methods. The methods are conversative, since the intercell flux are given by exact Riemann solvers, and the WENO limiters do not change the average of each conservative variable on each cell. The word “experiment” in our paper means running the solvers with different parameter (accuracy order, cell size, number of cores) combinations and compare the results with analytical solutions.
- The most obvious limitation is the mathematical model, the Euler equation of gas dynamics, which neglects viscosity. We will extend the methods in this paper to the Navier–Stokes equations in the future. The comparison between serial and parallel time costs has been made in the revised manuscript. The efficiency does not drop a lot but fluctuates a little in our tests. We have analyzed the reason of the relatively poorer efficiency of our solver compared with previous works. Each problem in this paper has been solved on multiple parameter combinations, and no instability has been observed.
- In the revised version, we have added some references to previous works in the section of results. Deeper discussions on parallel performance has been made, as stated in Response 3.
Round 2
Reviewer 2 Report
The authors have well described the methods now. It seems that Lazyweno is worse than Eigenweno for p=2, but for p=3, it seems to be the contrary
(but both are quite good for p=3; the difference is more visible for p=2). Do you agree with that and can you comment on it?
Author Response
Thanks for the comments.
In smooth regions, it's hard to tell which one is better. However, in regions near discontinuities, the curves given by LazyWeno are generally more oscillatory than those given by EigenWeno. This can be seen in the zoomed window in the revised figure. That's why we say "the EigenWeno limiter generally works better than its LazyWeno counterpart" at the end of that section.
Reviewer 3 Report
I think the authors have already accommodated the reviewer's suggestion, and the article can be accepted for publication.
Author Response
Thanks for the review.